# Standardization of laparoscopic trays using an inventory optimization model to produce immediate cost savings and efficiency gains

Jay Toor[1]◉*, Ajay Shah[1]◉, Aazad Abbas[2]◉, Jin Tong Du[2]◉, Erin Kennedy[1,3]

**1** Department of Surgery, University of Toronto, Toronto, Ontario, Canada, **2** Temerty Faculty of Medicine, University of Toronto, Toronto, Ontario, Canada, **3** Division of General Surgery, Sinai Health, Toronto, Ontario, Canada

◉ These authors contributed equally to this work.
* jaysinghtoor@gmail.com

**Data Availability Statement:** All relevant data are within the paper and its Supporting information files.

## Abstract

Perioperative services comprise a large portion of hospital budgets; the procurement and processing of surgical inventories can be an area for optimization in operational inefficiency. Surgical instrument trays can be customized as procedure-specific or standardized as trays that can be used in numerous procedure types. We conducted an interventional study to determine the cost savings from standardizing laparoscopic surgery instrument trays. A single-period inventory optimization model was used to determine the configuration of a standardized laparoscopic (SL) tray and its minimal stock quantity (MSQ). Utilization of instruments on the general surgery, gynecology, and gynecological oncology trays was recorded, and daily demand for trays (mean, SD) was assessed using daily operating room (OR) case lists. Pre- and post-intervention costs were evaluated by reviewing procurement data and quantifying medical device reprocessing (MDR) and OR processes. The SL tray was trialled in the OR to test clinical safety and user satisfaction. Prior to standardization, the customized trays had a total inventory size of 391 instruments (mean instruments per tray: 17, range: 12–22). Daily demand was an MSQ of 23 trays. This corresponded to a procurement cost of $322,160 and reprocessing cost of $41,725. The SL tray (mean instruments per tray: 15, mean trays/day: 9.2 ± 3.2) had an MSQ of 17 trays/day. The total inventory decreased to 255 instruments, corresponding to a procurement cost of $266,900 with savings of $55,260 and reprocessing cost of $41,562 with savings of $163/year. After 33 trial surgeries, user satisfaction improved from 50% to 97% (p < .05). Standardization to a single SL tray using the inventory optimization model led to increased efficiency, satisfaction, and significant savings through aggregating specific service demands. The inventory optimization model could provide custom solutions for various institutions with the potential for large-scale financial savings. Thus, future work using this model at different centres will be necessary to validate these results.

**Funding:** The author(s) received no specific funding for this work.

**Competing interests:** The authors have declared that no competing interests exist.

## Introduction

In the current healthcare environment, cost containment is more important than ever [1]. Perioperative services are often scrutinized as they comprise one-third of a hospital's budget [2]. The procurement and processing of surgical instruments make up a significant portion of these budgets [3]. Accordingly, increasing attention is being devoted to determining how to generate cost savings via process improvements and quality improvement initiatives. For example, a recent systematic review compared costs of disposable versus reusable surgical instruments, with inconclusive results due to a lack of primary data [4].

In particular, the focus has been placed on optimizing the surgical instrument tray by removing instruments used infrequently to reduce reprocessing costs. The techniques used include Lean-style clinician review, cost analysis and mathematical modelling [5–12]. Larger-scale optimization studies have been published in the industrial engineering literature, although significant limitations prevent practical application in the hospital setting, including institutional barriers to cultural change [5].

One possible strategy to reduce procurement and processing costs of surgical instruments is to consider the standardization of laparoscopic trays. Unlike the trays in specialties such as orthopedic surgery that are typically quite customized to a specific surgery type, laparoscopic instruments can be used by both general surgeons and gynecologic surgeons. However, no prior studies have explored the concept of combining trays from these surgical specialties into a standardized laparoscopic tray. We hypothesize that standardization to a single tray via demand aggregation may reduce the total inventory requirement and result in fewer instrument purchasing, processing and replacement costs. Secondary outcomes include perioperative staff satisfaction.

## Methods

### Study design

This was a single-site interventional quality improvement study conducted at a large academic hospital in a major metropolitan city which performs over 800 general surgery (GS), gynecology (GY), and gynecological oncology (GO) procedures annually using customized specialty-specific trays.

The minimum stock quantity (MSQ) of the GS, GY, and GO trays were calculated using a single-period inventory optimization model (IOM) based on the Newsvendor problem [13].

Briefly, the Newsvendor problem determines the optimal inventory required for a given perishable item. The classic analogy for this problem is that of a newspaper vendor who needs to know how many copies of the day's paper to stock while faced with varying demand. Since items on surgical trays need to be reprocessed after use, this model is directly applicable to this study. Accordingly, the MSQ can be modelled by the following equation:

$$MSQ = \mu + z \cdot \sigma, \tag{1}$$

where $\mu$ is the mean utilization of the trays, $z$ is the service level of the trays, and $\sigma$ is the standard deviation of utilization. The trays were then standardized into a single standardized laparoscopic (SL) tray, and its MSQ was calculated.

The effect of standardization was measured by determining the difference in procurement costs, tray reprocessing costs, and total reprocessing costs for the customized trays versus the SL tray. This study was performed with the assistance of the managers of the Operating Room (OR) and Medical Device Reprocessing Department (MDRD).

## Optimization of SL tray

All instruments used in three customized trays were combined and standardized into a single SL tray. This was completed through a clinician review of tray contents by all users and a mathematical model based on observations of intraoperative instrument utilization. The methodology was described in detail in the literature [13]. The SL tray was presented to laparoscopic surgeons, registered nurses (RNs) and medical devices reprocessing (MDR) technicians. It was thenceforth utilized in place of the previously customized trays in all subsequent laparoscopic procedures.

## Data collection

Hospital procedure data for the preceding four fiscal years (2014–2018) was retrieved. Procurement and reprocessing costs were provided by the manager of the MDRD. The number of daily GS, GY and GO procedures over this period was calculated to estimate the number of customized trays used. An observer was present in the OR during GS, GY, and GO procedures to document instrument use over a period of 3 weeks; variables collected were: procedure type, surgeon name, instruments used, pick list contents, pick list related adverse events, trayy related adverse events., and intra-operative item retrieval time. This observer was also present in the MDRD to document tray reprocessing; variables collected were: instrument name, time to decontaminate, time to assemble, total tray assembly time, and list assembly time [13].

## Minimal Stock Quantity (MSQ)

The daily number of procedures during the four-year period was used to determine the MSQ requirement for customized and standardized trays. The general demand required inventory of the UL tray ($MSQ_{UL}$) for a set of trays $T$ of size $N$ may be described as follows:

$$MSQ_{UL} = \sum_{i=1}^{N} \mu_i + z \cdot \sqrt{\sum_{i=1}^{N} \sigma_i^2}, \tag{2}$$

where $i$ is a tray in the set $T$, $\mu_i$ is the mean demand of each tray, and $\sigma_i$ is the standard deviation of tray $i$. Simplifying for our case of GS, GY, and GO trays, we have:

$$MSQ_{ST} = (\mu_{GS} + \mu_{GY} + \mu_{GO}) + z \cdot \sqrt{\sigma_{GS}^2 + \sigma_{GY}^2 + \sigma_{GO}^2} \tag{3}$$

Since we cannot have a fraction of a surgical tray, the MSQ is always rounded up to the nearest integer to ensure there is safety stock available.

## Cyclic service level assumption

The cyclic service level (CSL) is defined as the probability that a tray is available when it is needed for a procedure [14]. We chose a CSL of 99%; this infers that only 1 out of 100 procedures will not have a tray available when it is requested. This is an essential component of the equations required to calculate MSQ. Accordingly, the CSL can be modelled by the following equation:

$$CSL = \frac{c_{overage}}{c_{overage} + c_{underage}}, \tag{4}$$

$c_{overage}$ is the overage cost (cost of having an extra tray in inventory when it is not needed) and $c_{underage}$ is the underage cost (cost of not having a tray available when it is needed i.e., stockout event). In the healthcare setting, it is difficult to place a discrete financial value on a stockout

event as the implications extend beyond the quantifiable loss of OR time into the domain of impact on patient care.

## Cost metrics

The procurement costs for each surgical instrument were obtained from previous hospital invoices. The procurement costs for each surgical tray were calculated as the sum of the procurement costs for each instrument on the tray. The total procurement costs for each tray were calculated as the product of the procurement costs for each tray and the MSQ for each instrument. Procurement cost ($C_{procurement}$) may be defined as the costs associated with obtaining the surgical trays. For some surgical tray $T$ with $N$ number of instruments, $C_{procurement}$ may be defined as:

$$C_{procurement} = MSQ \cdot \sum_{i=1}^{N} c_i, \tag{5}$$

where $c_i$ is the procurement cost for each instrument on the surgical tray, and $MSQ$ is the minimum stock quantity of the surgical tray.

The reprocessing costs for each instrument on the tray were determined via timed observation of instrument reprocessing. The mean reprocessing time for each instrument on each tray was calculated, and the reprocessing costs for each tray was determined as the sum of the mean reprocessing costs of all instruments on the tray. As such, the reprocessing costs ($C_{reprocess}$) can be defined as:

$$C_{reprocess} = t_{reprocess} \cdot W, \tag{6}$$

where $t_{reprocess}$ is reprocessing time in minutes, and $W$ is the MDR technician wages per minute in CAD. The total reprocessing costs per year for each tray would be determined as a product of the reprocessing costs for each tray and the yearly number of procedures requiring each tray. Reprocessing costs are calculated annually. The difference in procurement and reprocessing costs between the customized trays and the SL tray were used to determine the overall cost savings obtained with the standardized tray model. The difference was calculated using a side-by-side cost comparison.

## User satisfaction and safety

Pre- and post-intervention surveys were distributed to surgeons, RNs and MDR technicians comparing satisfaction between customized and standardized inventory. The questionnaire was administered at two time points: immediately before tray standardization and one month after. Respondents were asked if they were (1) satisfied or (2) dissatisfied with the inventory configuration. For surgeons, satisfaction refers to the ability to perform a given procedure without concerns regarding the availability of required instruments. For RNs and MDR technicians, satisfaction was referred to the requirements of familiarity with instrument and tray configurations. The chi-squared statistical test was used to analyse the satisfaction survey results, with significance set at $p < .05$.

# Results

## Customized trays (GS, GY, GO)

The variables used in the mathematical model are shown in Table 1. Prior to standardization, there were three procedure-specific tray types: GS, GY, and GO, with a total inventory size of 391 instruments (mean instruments per tray: 17, range: 12–22). Daily demand was calculated for GS (mean 4.6 trays/day, SD ±2.1), GY (2.8 ±2.1), and GO (1.8 ±1.2) trays, with an MSQ of

**Table 1. Variables for Newsvendor formula used to determine MSQ per tray.**

| Tray Name | Demand (Trays per day) | SD of Demand (Trays per day) | Cyclic Service Level (%) | Z Score | Raw Minimal Stock Quantity (MSQ) | Rounded Minimal Stock Quantity (MSQ) |
|---|---|---|---|---|---|---|
| GS | 4.6 | 2.1 | 99% | 2.33 | 9.49 | 10 |
| GY | 2.8 | 2.1 | 99% | 2.33 | 7.69 | 8 |
| GO | 1.8 | 1.2 | 99% | 2.33 | 4.59 | 5 |
| SL | 9.2 | 3.2 | 99% | 2.33 | 16.65 | 17 |

GS: general surgery, GY: gynecology, GO: gynecology oncology, SL tray: standardized laparoscopic tray.

**Table 2. Costs savings breakdown per tray.**

| Tray Name | Procurement Cost per Tray (CAD) | Number of Instruments | Minimal stock quantity (MSQ) | Total Procurement Cost for Necessary Inventory (CAD) | Reprocessing Labour Costs* /Tray (CAD) | Procedures per tray | Total Reprocessing Costs (CAD) |
|---|---|---|---|---|---|---|---|
| GS | $12,360 | 22 | 10 | $123,600 | $14.46 | 1548 | $22,415.05 |
| GY | $15,720 | 12 | 8 | $125,760 | $17.59 | 898 | $15,795.82 |
| GO | $14,560 | 15 | 5 | $72,800 | $10.88 | 323 | $3,514.24 |
| SLT | $15,700 | 15 | 17 | $266,900 | $15.01 | 2769 | $41,562.69 |

GS: general surgery, GY: gynecology, GO: gynecology oncology, SL: standardized laparoscopic.

23 trays total. This corresponded to a total procurement cost of $322,160 and reprocessing cost of $41,725, as shown in Table 2. The configurations of the various trays used may be found in Table 3. The configuration of specific instruments in each customized tray can be found in the S1-S3 Tables in S1 File.

## Standardized laparoscopic tray

An SL tray was generated, with instruments listed in the S4 Table in S1 File. The SL tray had lower daily tray and instrument requirements (mean instruments per tray: 15, mean trays/day: 9.2 ±3.2). This reduced the MSQ by approximately 26% (MSQ: 17 trays). After standardization,

**Table 3. Configuration of the various trays used in this study.**

| General Surgery Tray | Gynecology Tray | Gynecology Oncology Tray | Standardized Laparoscopic Tray |
|---|---|---|---|
| Dissector (1) | Rounded/bullet graspers (2) | Small tip laparoscopic clinch (2) | Bowel grasper ratchet (2) |
| Debakey flat (2) | Fundus grasper (1) | Hunter grasper (2) | Bowel grasper non (1) |
| Crile (1) | Needle nose grasper (1) | Duckbill grasping forceps (2) | Maryland dissector (2) |
| Horizontal clip applier (1) | Maryland dissector (1) | Maryland dissector (1) | Rounded grasper (2) |
| Suction irrigator (4) | Suction irrigator (1) | Suction irrigator (1) | Suction irrigator (1) |
| Bowel grasper (3) | L hook (1) | L hook (1) | L hook (1) |
| Laparoscopic scissors (1) | Probe (1) | Laparoscopic scissors (1) | Horizon clip applier (1) |
| L hook (1) | Bipolar forceps (1) | Needle driver (1) | Reusable laparoscopic shears/scissors (1) |
| Claw forceps (1) | Bowel grasper (3) | Maryland dissector (1) | Laparoscopic Kocher (1) |
| Senn retractors (2) | Maryland dissector (1) | | Needle driver (1) |
| Right angle retractor (2) | Needle driver (2) | | Bipolar forceps (1) |
| Maryland dissector (1) | | | |
| Needle driver (2) | | | |

GS: general surgery, GY: gynecology, GO: gynecology oncology, SL: standardized laparoscopic. Numbers in brackets indicate the number of each instrument.

**Table 4. Satisfaction survey results of customized trays vs standardized trays.**

| Staff | Satisfied (n) | Unsatisfied (n) |
|---|---|---|
| **Customized Inventory** | | |
| Surgeons | 7 | 1 |
| RNs and MDRs | 8 | 14 |
| Total | 15 | 15 |
| **Standardized Inventory** | | |
| Surgeons | 8 | 0 |
| RNs and MDRs | 21 | 1 |
| Total | 29 | 0 |

the total inventory decreased to 255, corresponding to a procurement cost of $266,900 (savings of $55,260) and reprocessing costs of $41,562 (savings of $163/year).

## User satisfaction and safety

30 staff completed the pre- and post-intervention satisfaction questionnaire (8 surgeons, 12 RNs, and 10 MDR technicians) at 1-month pre-standardization and 1-month post-standardization, respectively. The proportion of all staff satisfied with instrument configuration was 50% with the customized inventory and 97% with the standardized inventory. The satisfaction for surgeons was not significantly different between the customized and standardized trays ($\chi^2 = 0.39$, p = .53). The satisfaction of RNs and MDR technicians between the customized and standardized trays was significantly different ($\chi^2 = 17.1$, p < .05). Table 4 includes results of the satisfaction survey. RNs and MDR technicians were unsatisfied with the customized trays due to redundant infrequently used instruments and missing critical instruments.

## Discussion

The most important finding in this study is that the conversion of customized specialty-specific trays to a single SL tray resulted in significant annual cost savings via reduced total instrument inventory and tray quantity requirements to meet daily demand. This study showed that the use of mathematical optimization is a successful strategy to realize savings in the procurement of surgical instruments in the OR setting. The practical and quality improvement aspects of implementing this intervention are described in Toor et al. [13]. As expected, reprocessing and instrument turnover costs stayed relatively unchanged as the number of procedures per year primarily dictates the reprocessing costs [15].

The financial benefits of standardization versus customization of instrument trays are well documented [13], which suggests that this mathematical model may be generalizable for laparoscopic tray standardization at other hospitals [2]. In this study, the subspecialty trays had already undergone optimization before standardization, thereby under-estimating the potential cost savings. Additionally, the minimum quantity required was for 99% quantity satisfaction, higher than the 96% currently required in hospitals, which further underestimates real-world savings. This mathematical inventory optimization process has shown promising financial and clinical satisfaction results with other specialty trays, including large Orthopaedic and ENT trays [13]. The hybrid model of mathematical and clinician-directed optimization will likely mitigate the shortcomings of using either approach alone.

During the process of tray standardization, several factors were identified that could influence the degree of cost savings from similar interventions at other institutions. These broadly fall into surgical scheduling, inventory procurement, and workflow efficiency.

## Surgical schedule variability's impact on inventory

The calculation of MSQ shown in Eq (1) reveals an important consideration when scheduling surgeries over multiple days. For example, in a scenario with a daily demand of 5 trays (SD 3), an MSQ of 12 is required to maintain a 99% service level. In contrast, reducing SD to 2 reduces MSQ to 10. Increasing the mean to 6 (SD 2) only requires an MSQ of 11. This demonstrates the disproportionately large effect of schedule variability on MSQ. A surgical schedule that reduces variability by scheduling the same number of surgeries daily will reduce SSQ and MSQ. If tray demand variability is low such as in a "smooth" surgical schedule, then the benefits of demand aggregation to an SL tray are reduced.

## Volume-based procurement discounts

Procurement cost decreases can occur from bulk purchases made possible by standardization. For example, at a 15% discount, savings would increase from $55,260 to $95,295 per annum. In our case, we did not receive a direct financial discount for procurements during the study period.

## The effect of standardization on the workforce

A common argument in favour of inventory standardization is the reduction in various instruments and tray types. This reduces task variety for nurses and MDR technicians, who will have only a single laparoscopic tray to be familiar with during the tasks of OR case set-up and instrument reprocessing, respectively. A decrease in task variety has long been understood to increase efficiency and decrease time spent per task [16, 17]. Increased satisfaction rates among MDR technicians and RNs support this assumption.

## Limitations

The primary limitation of this study is the inability to assess all impacts of the intervention comprehensively. For example, we could not calculate the number of instruments opened in peel-packs after the intervention. The implications of this intervention at a large time scale (e.g., tray/instrument maintenance, instrument replacement) escape the scope of this study. Apart from no explicit adverse safety incidents being documented in the post-intervention period, there is no way to conclusively determine that this intervention had no negative impact on patient safety. In addition, the clinician review was conducted over a short period. It consisted of only 30 staff, which may have revealed issues such as missing instruments for all procedures and all surgeons. However, given the cost savings and high satisfaction, it is likely that patient safety was not impaired.

Furthermore, this study hinges on the ability of laparoscopic trays to be used by many specialties such as GS, GY, and GO, allowing the creation of a single SLT that can serve different specialties. The concept of standardization may be more challenging to apply to other trays, such as the Orthopedic Surgery trays, which are more differentiated for that specific surgical specialty. Thus different optimization methods will have to be considered [13]. Lastly, a normal distribution was used to approximate the probability distribution of daily demand on surgical trays. Given that our sample size was procedures per day over two years, our large sample size justifies this assumption as suggested by the Central Limit Theorem [18]. Although there are limitations to the generalized application of standardization over customization, this underscores the utmost importance of a systematic, mathematically sound approach to decision-making. The limitations above and discussions reinforce the importance of our

mathematical model; it creates a custom solution for each institution's unique circumstances with the potential for large-scale financial savings.

## Conclusion

Standardizing three laparoscopic surgical trays (general surgery, gynecology, gynecological oncology) into a single standardized laparoscopic tray resulted in significant cost savings and increased satisfaction at our institution. However, as illustrated by our study, the potential cost impacts may differ between institutions, underscoring the importance of using an evidence-based business approach such as the inventory optimization model. Future work should involve multiple centers to address generalizability.

## Supporting information

**S1 File. Contains all the supporting tables and figures.**
(DOCX)

**S1 Data.**
(XLSX)

## Acknowledgments

We thank all the surgeons, nurses, and MDR technicians who made this research possible.

## Author Contributions

**Conceptualization:** Jay Toor, Ajay Shah, Aazad Abbas, Jin Tong Du.

**Data curation:** Jay Toor, Ajay Shah, Aazad Abbas, Jin Tong Du.

**Formal analysis:** Jay Toor, Ajay Shah, Aazad Abbas, Jin Tong Du.

**Funding acquisition:** Erin Kennedy.

**Investigation:** Jay Toor, Ajay Shah, Aazad Abbas, Jin Tong Du.

**Methodology:** Jay Toor, Ajay Shah, Aazad Abbas, Jin Tong Du.

**Project administration:** Jay Toor, Ajay Shah, Aazad Abbas, Jin Tong Du, Erin Kennedy.

**Resources:** Erin Kennedy.

**Supervision:** Erin Kennedy.

**Writing – original draft:** Jay Toor, Ajay Shah, Aazad Abbas, Jin Tong Du.

**Writing – review & editing:** Jay Toor, Ajay Shah, Aazad Abbas, Jin Tong Du, Erin Kennedy.

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
