## [Decision Letter · Decision Letter 0]

24 Jan 2022

PONE-D-21-17028Standardization of laparoscopic trays using an inventory optimization model to produce immediate cost savings and efficiency gainsPLOS ONE

Dear Dr. Toor,

Thank you for submitting your manuscript to PLOS ONE. After careful consideration, we feel that it has merit but does not fully meet PLOS ONE’s publication criteria as it currently stands. Therefore, we invite you to submit a revised version of the manuscript that addresses the points raised during the review process.

We look forward to receiving your revised manuscript.

Kind regards,

Federico Ferrari

Academic Editor

PLOS ONE

Journal Requirements:

Furthermore, we have noted that ethics approval has been waived by Mount Sinai Hospital Research Ethics Board for this particular study. Please ensure that this is indicated in the manuscript text.

Reviewers' comments:

Reviewer's Responses to Questions

**Comments to the Author**

1. Is the manuscript technically sound, and do the data support the conclusions?

Reviewer #1: Yes

Reviewer #2: Yes

2. Has the statistical analysis been performed appropriately and rigorously? 

Reviewer #1: Yes

Reviewer #2: Yes

3. Have the authors made all data underlying the findings in their manuscript fully available?

Reviewer #1: Yes

Reviewer #2: Yes

4. Is the manuscript presented in an intelligible fashion and written in standard English?

Reviewer #1: Yes

Reviewer #2: Yes

5. Review Comments to the Author

Reviewer #1: This article has standard language and clear thinking.This study mainly demostrated that Standardization of three laparoscopic surgical trays (GS, GY, GO) into a single SL tray resulted in 261 significant cost savings and increased satisfaction at our institution. I have 2 suggestions for this study. First,the statistical method is not very clear in this article.The statisical method using in every test should be clearly described.Next,The actual clinical application effect in different regions needs to be further analyzed by increasing the number of samples and adopting multi-center method.

Reviewer #2: This scientific paper is vell written and informed. The content of study is very interesting and useful for surgeons, RNs ann MDRs. I congratualte the author for their successful research which will play a very important role for cost control in operating room. There are some grammatical and typing errors in this manuscript.

6. PLOS authors have the option to publish the peer review history of their article (what does this mean?). If published, this will include your full peer review and any attached files.

Reviewer #1: **Yes: **kai Zhang

Reviewer #2: **Yes: **Tuerhongjiang Tuxun

---

## [Author Response · Author response to Decision Letter 0]

7 Apr 2022

Dear Drs. Ferrari and Chenette, 

 Thank you on behalf of the reviewers for your consideration of our manuscript “Standardization of laparoscopic trays using an inventory optimization model to produce immediate cost savings and efficiency gains” to be considered for publication in PLOS ONE. We have taken the time to consider the feedback from the reviewers, and we have made the following changes to the manuscript:

Reviewer #1: 

“First, the statistical method is not very clear in this article” Thank you kindly for this comment. We elaborated that we used a side-by-side cost comparison (Line 149-150). We have also described our methodology in detail including equations. Having incorporated the suggestion, we are confident the readers of PLOS ONE will be able to better understand our work.

“The actual clinical application effect in different regions needs to be further analyzed by increasing the number of samples and adopting multi-center method” Thank you for this insightful comment. We included the suggestion in Line 266-267.

Reviewer #2: 

“There are some grammatical and typing errors in this manuscript.” Thank you for this constructive feedback. We have reviewed the manuscript in great detail and corrected any grammar and punctuation errors (Line 50, 75). 

Thank you for your time and efforts in enhancing our work.

Sincerely, 

Jay Toor MD MBA

Orthopaedic Surgery Resident Physician

Department of Surgery

University of Toronto

Tel: 1-416-918-9519

Email: jaysinghtoor@gmail.com

---

## [Decision Letter · Decision Letter 1]

6 Jul 2022

PONE-D-21-17028R1Standardization of laparoscopic trays using an inventory optimization model to produce immediate cost savings and efficiency gainsPLOS ONE

Dear Dr. Toor,

Thank you for submitting your manuscript to PLOS ONE. After careful consideration, we feel that it has merit but does not fully meet PLOS ONE’s publication criteria as it currently stands. Therefore, we invite you to submit a revised version of the manuscript that addresses the points raised during the review process.

We look forward to receiving your revised manuscript.

Kind regards,

Kush Raj Lohani, Master of Surgery

Academic Editor

PLOS ONE

Reviewers' comments:

Reviewer's Responses to Questions

**Comments to the Author**

1. If the authors have adequately addressed your comments raised in a previous round of review and you feel that this manuscript is now acceptable for publication, you may indicate that here to bypass the “Comments to the Author” section, enter your conflict of interest statement in the “Confidential to Editor” section, and submit your "Accept" recommendation.

Reviewer #2: All comments have been addressed

Reviewer #3: (No Response)

2. Is the manuscript technically sound, and do the data support the conclusions?

Reviewer #2: (No Response)

Reviewer #3: Partly

3. Has the statistical analysis been performed appropriately and rigorously? 

Reviewer #2: (No Response)

Reviewer #3: Yes

4. Have the authors made all data underlying the findings in their manuscript fully available?

Reviewer #2: (No Response)

Reviewer #3: Yes

5. Is the manuscript presented in an intelligible fashion and written in standard English?

Reviewer #2: (No Response)

Reviewer #3: Yes

6. Review Comments to the Author

Reviewer #2: (No Response)

Reviewer #3: Thank you for inviting me to review this paper.

The manuscript reports a study that aimed to produce cost and efficiency gains by standardizing the contents of laparoscopic surgery instrument trays. Research and improvement in the delivery of sterile processing services is an important and often overlooked component of surgical safety and efficiency. While there have been a small number of similar studies this can be a useful addition to that literature.

It seems that there have been prior reviews, so it’s unfortunate, and probably frustrating for the authors, that I have a range of new comments; however, while challenging, I feel they are important and would substantially enhance the paper.

Overall, the focus on mathematical approaches belies some of the nuanced complexities of sterile processing. There are also quite a few areas where more clarification or precision is warranted (e.g. survey response rates; costs ‘per year’ or other). I also don’t feel that the operational consequences have been fully thought through. There is no discussion of how the trays were changed and the new “standard” was introduced into practice.

Looking at the tables detailing the changes, one original tray has more instruments than the final one. Does this mean that the new tray was missing critical instruments for certain procedures, or simply the older tray had legacy instruments that weren’t used (in which case this isn’t just about standardization but also optimization)? Similarly, another original tray has exactly the same number of instruments as the new tray – are these the same or are there differences? Definitely missing here is the consideration – at least a discussion point - not just of the numbers of instruments in a tray, but also the types and uses of instruments.

I think it’s really valuable work, but can’t help thinking it’s ignored many of the real-world implications that may make this appear to be more straightforward that in actually is. Addressing the wider context of the work system in the Introduction and Discussion would make this a much stronger and more relevant paper.

INTRO

The first sentence of a paper or abstract is important. Consider qualifying why surgical inventories can be a “source of inefficiency” with some reasons why and / or some effect. Of course, they can also be a source of efficiency.

The introduction is very short. There is space to expand on the learning from prior work. I’d like to see more on the reasons why there are “limitations preventing practical application.” You could also discuss a little more the methods and results of the other prior studies, especially as you haven’t addressed this in your own work. An extra sentence or 3 would be valuable for establishing how your study builds on this prior work.

The introduction doesn’t really go into the challenges or trade-offs of standardization. Including surgeon preferences, generic vs specific uses, or the trade-off between having a range of instruments to cover a broader set of preferences/procedures, which may lead to more instruments in a tray vs specialist trays with less instruments. It might not need that, but I think a little more nuanced thought about the challenges of “standardization” and what it really means would be helpful. Of course, standardization can lead to standardized inefficiency.

In moving on to the methods, I would also like to see more discussion of the background of the “Newsvendor” problem, and the modelling methods chosen. Most readers will be unfamiliar with these. This could be in the intro or methods, but I would suggest the former as that is naturually more discursive.

METHODS

How were procurement & reprocessing costs discovered?

What did the “clinical review” of the SL tray involve? How many clinicians? What did they review? Were any operational trials (ie. A clinical review might not reveal missing instruments for all procedures & all surgeons).

How might this affect scheduling? If you have different procedure-specific trays, simultaneous surgeries would not be an issue. However, instead with “one size fits all” trays you presumably need more of them, or carefully schedule procedures and reprocessing to allow appropriate sharing?

Might using “mean” demand not immediately disadvantage periods of demand above the mean? Ie. During peak times, trays might not be available if calculations are only done on “means”. I think you allude to this later, but a little more about it in the methods could be useful.

Did you study or explore the types and uses of instruments in each tray? How did you know it was acceptable to remove some and not others? Are any of your team actively involved in sterile processing, or did you conduct any site vists?

How did you make the changes? Did you do this over time or change all the trays over to the standardized ones in one go? How did you ensure that no critical instruments were missing? Some sense or narrative of how you did this would be very informative.

Where and how were timed observations of instrument reprocessing made? Did this time include decontamination, and assembly? In assembly, did you study the time taken for inspection, functional checking and missing instrument retrieval (as well as ensuring all instruments are present)?

Mean instruments per tray of 17 seems relatively low compared to some trays. Do you think this has implications for generalization of your methods/findings to other trays?

Are the calculated reprocessing costs yearly, monthly, or per procedure? Time scale would be precise here.

RESULTS

How can the GS tray have more instruments than the standardized tray? Does this mean that there were redundant instruments in the GS tray, or that 7 extra instruments were needed for any GS procedure? That would make the cost saving calculations misleading. What was the overlap in instruments between the “old” and “new” versions (i.e. which stayed the same, were added, or removed?). Maybe that’s all too much, but the numbers don’t tell us anything about how the new tray differed. That’s really important.

Presumably the “standardized” trays basically used the same instruments – so the 15 in GO wer the same 15 in SLT….so aren’t GO and SLT the same tray? If not, presumable in the SLT tray there were instruments that were needed in the GO cases (so important instruments were missing from the SLT); or that some of the instruments in the GO trays were never used (which means this is less about standardization and more about just removing unused instruments).

There was a lot of dissatisfaction with the customized trays with RNs and MDRs. Why was this?

How long after the standardization process was the survey delivered? How was it delivered (onoine,paper etc)? How many people was it sent to vs completed it (% response rate?).

I could not see any data on the reprocessing times for each instrument (as per the method).

DISCUSSION

The mathematical focus of this paper ignores many of the operational nuances that may be found in this type of approach.

“Financial benefits….well documented” – this is a very broad brush. In fact, it’s well documented that the wrong sort of standardization can lead to inefficiencies. Furthermore, standardization and patient-centered care are often in opposition.

How do you think your approach would fare with other types of trays e.g. general surgery? These trays often have >100 instruments. What about the trade-offs with

How many surgeries required extra instruments (or “peel packs?”)?

One challenge with standard trays is that surgeons who trained differently have different preferences. Did you find that this suited some surgeons more than others; that they all used similar approaches; or that the particular procedures you worked on required a standard set of instruments, with little personal or procedural variation?

The cost calculations are well discussed but I think this misses a lot of nuanced implications / discussion points.

Did you calculate the “lifespan” of the instruments? Presumably if they are being turned over faster, they will need to be replaced sooner?

Again, please state over what time period the “savings” (line 220) refer to,

Some trays have rarely used instruments that nevertheless are important to have at the right time. How/are you sure that you didn’t remove instruments like this?

How did the implementation of the new trays happen? Making change in organizations is complex, especially with surgeons who are used to doing this a certain way. Did this present challenges?

Overall, the conclusion should consider not just the mathematical calculations, but the safety, quality, and operational implications for this work. Can this approach be used on all trays or only a subset? Are the authors planning more studies in other specialties or procedures or trays?

As I’ve said, there’s no consideration (as far as I can tell) for the implications in tray assembly, routine maintenance, or replacement of instruments.

7. PLOS authors have the option to publish the peer review history of their article (what does this mean?). If published, this will include your full peer review and any attached files.

Reviewer #2: **Yes: **Tuerhongjiang Tuxun

Reviewer #3: No

---

## [Author Response · Author response to Decision Letter 1]

14 Aug 2022

Dear Dr. Ferrari, 

 Thank you on behalf of the reviewers for your re-consideration of our manuscript “Standardization of laparoscopic trays using an inventory optimization model to produce immediate cost savings and efficiency gains” to be considered for publication in PLOS ONE. We have taken the time to consider the most recent feedback from the reviewers, and we have made the following changes to the manuscript, submitted in tracked changes and clean versions:

Overall, the focus on mathematical approaches belies some of the nuanced complexities of sterile processing. There are also quite a few areas where more clarification or precision is warranted (e.g. survey response rates; costs ‘per year’ or other). I also don’t feel that the operational consequences have been fully thought through. There is no discussion of how the trays were changed and the new “standard” was introduced into practice.

Looking at the tables detailing the changes, one original tray has more instruments than the final one. Does this mean that the new tray was missing critical instruments for certain procedures, or simply the older tray had legacy instruments that weren’t used (in which case this isn’t just about standardization but also optimization)? Similarly, another original tray has exactly the same number of instruments as the new tray – are these the same or are there differences? Definitely missing here is the consideration – at least a discussion point - not just of the numbers of instruments in a tray, but also the types and uses of instruments.

I think it’s really valuable work, but can’t help thinking it’s ignored many of the real-world implications that may make this appear to be more straightforward that in actually is. Addressing the wider context of the work system in the Introduction and Discussion would make this a much stronger and more relevant paper.

Thank you for these helpful comments. The purpose of this paper was to describe the process of developing a tray that was both mathematically and functionally optimized to serve as a standardization of three existing trays for laparoscopic surgeries. We have included citations to other publications which describe the quality improvement initiatives that were undertaken to operationalize these changes. We have also included more details as to the specific types of instruments that were removed/included, which have also been published in another paper. We thank you for your helpful review, and hope that you find our responses below satisfactory.

INTRO

The first sentence of a paper or abstract is important. Consider qualifying why surgical inventories can be a “source of inefficiency” with some reasons why and / or some effect. Of course, they can also be a source of efficiency.

Thank you for this comment. The first line of the abstract was amended to explain why this domain was chosen as a target for operational efficiency optimization. (Lines 27-28).

The introduction is very short. There is space to expand on the learning from prior work. I’d like to see more on the reasons why there are “limitations preventing practical application.” You could also discuss a little more the methods and results of the other prior studies, especially as you haven’t addressed this in your own work. An extra sentence or 3 would be valuable for establishing how your study builds on this prior work.

Thank you for this comment. Unfortunately, the literature around surgical inventory optimization is lacking, which is noted in the discussion. We have added some context by providing a comment on the barriers to practical application of optimization, and some of the other surgical inventory research. (Lines 58-60 and 65-66).

The introduction doesn’t really go into the challenges or trade-offs of standardization. Including surgeon preferences, generic vs specific uses, or the trade-off between having a range of instruments to cover a broader set of preferences/procedures, which may lead to more instruments in a tray vs specialist trays with less instruments. It might not need that, but I think a little more nuanced thought about the challenges of “standardization” and what it really means would be helpful. Of course, standardization can lead to standardized inefficiency.

Thank you for this helpful comment. While we agree that this discussion is within the scope of this paper, we have elected to carry out an analysis of this topic in the Discussion (Lines 255-260)

In moving on to the methods, I would also like to see more discussion of the background of the “Newsvendor” problem, and the modelling methods chosen. Most readers will be unfamiliar with these. This could be in the intro or methods, but I would suggest the former as that is naturually more discursive.

Thank you for this comment. We agree with your suggestion, and will include background about the Newsvendor problem in the Methods (Lines 84-87).

METHODS

How were procurement & reprocessing costs discovered? What did the “clinical review” of the SL tray involve? How many clinicians? What did they review? Were any operational trials (ie. A clinical review might not reveal missing instruments for all procedures & all surgeons).

Thank you for this comment. Procurement and reprocessing costs were provided by the manager of the MDRD, and we have clarified in Line 108-109. As described in the results section under “User satisfaction and safety”, clinician review involved pre-and post- intervention satisfaction questionnaire. A total of 30 clinicians including 8 surgeons, 12 RNs, and 10 MDR technicians completed the survey measuring their satisfaction with the traditional and standardized inventory. We agree that this clinician review may not review missing instruments for all procedures and all surgeons and have included this as a limitation. 

How might this affect scheduling? If you have different procedure-specific trays, simultaneous surgeries would not be an issue. However, instead with “one size fits all” trays you presumably need more of them, or carefully schedule procedures and reprocessing to allow appropriate sharing?

Thank you for this comment. We understand the concern with scheduling. This was considered in calculating the MSQ (minimum stock quantity) which accounts for the general demand for required inventory. As shown by the results, the total stock quantity for GS+GY+GO pre-standardization was 10+8+5=23 trays, and post-standardization, 17 trays. As seen in this case, there is more SLT than any of the GS/GY/GO trays individually, however the standardization processed allowed us to decrease the number of total tray needed in stock. 

Might using “mean” demand not immediately disadvantage periods of demand above the mean? Ie. During peak times, trays might not be available if calculations are only done on “means”. I think you allude to this later, but a little more about it in the methods could be useful.

Thank you for this comment. The MSQ not only takes into consideration of the mean demand but as well as the variance in the demand as shown in equation 2 (Line 122), this is essentially the upper bound the number of tray needed routinely (greater than the mean). In addition, another assumption that is made is shown in the cyclic service level assumption (Line 131), which in our study was chosen to be 99%, inferring only 1/100 procedures will not have a tray available.

Did you study or explore the types and uses of instruments in each tray? How did you know it was acceptable to remove some and not others? Are any of your team actively involved in sterile processing, or did you conduct any site vists?

Thank you for this comment. Multiple trained observers were involved in the study by being present in the OR to document instrument usage and being present in the MDRD to document the tray reprocessing process. We have included this information in Line 111. One team member who was not an author on this specific paper was the manager of the MDRD, however, authors JT and JTD conducted many site visits observing both in the OR and MDRD. 

How did you make the changes? Did you do this over time or change all the trays over to the standardized ones in one go? How did you ensure that no critical instruments were missing? Some sense or narrative of how you did this would be very informative.

Thank you for this comment. The change to SLT occurred as a single event - we first collected the data from the observers and performed the analysis behind the scenes, then produced a SLT that was then used and tested. We were able to identify that there were no critical instruments missing given the positive user satisfaction. We agree that this is a limitation to ensuring that there will be no incidence of missing critical instruments in the future and have included this point in the limitations. 

Where and how were timed observations of instrument reprocessing made? Did this time include decontamination, and assembly? In assembly, did you study the time taken for inspection, functional checking and missing instrument retrieval (as well as ensuring all instruments are present)? 

Thank you for your comment. The timed observation of instrument reprocessing was made in the MDRD and by a trained observer. The variables collected have been added to the methods (Line 153-155). This did include the time for decontamination and assembly (including inspection and functional checking). Missing instrument retrieval was a variable collected in the OR observations. 

Mean instruments per tray of 17 seems relatively low compared to some trays. Do you think this has implications for generalization of your methods/findings to other trays? 

Thank you for your comment. There were 15 instruments on the SLT, which we agree is a decrease from the GS tray at 22 instruments. However, given the high user satisfaction with the SLT, it proved that if not all, at least most of the instruments that were removed from the traditional tray were not often used. In terms of the implication for generalization of our method to standardize trays - the laparoscopic tray is unique in that the majority of the instruments on the tray is used by all 3 specialties GS/GY/GO, and thus allowing us to create 1 tray. This is admittedly more difficult to do with other trays such as Orthopedics trays and we have included this as a limitation (Lines 274-277), however we do have another paper describing alternative optimization processes (Ref #13)

Are the calculated reprocessing costs yearly, monthly, or per procedure? Time scale would be precise here. 

Thank you for this comment. Reprocessing costs are calculated annually. We have included that in Line 160.

RESULTS

How can the GS tray have more instruments than the standardized tray? Does this mean that there were redundant instruments in the GS tray, or that 7 extra instruments were needed for any GS procedure? That would make the cost saving calculations misleading. What was the overlap in instruments between the “old” and “new” versions (i.e. which stayed the same, were added, or removed?). Maybe that’s all too much, but the numbers don’t tell us anything about how the new tray differed. That’s really important. 

Thank you for your comment. This is an excellent observation. There were unnecessary legacy instruments on the GS tray that were removed to form the SLT. A configuration of the trays used in the study can be found in Table 3 which demonstrate the overlap between the traditional GS, GY, GO trays and the SLT. 

Presumably the “standardized” trays basically used the same instruments – so the 15 in GO wer the same 15 in SLT….so aren’t GO and SLT the same tray? If not, presumable in the SLT tray there were instruments that were needed in the GO cases (so important instruments were missing from the SLT); or that some of the instruments in the GO trays were never used (which means this is less about standardization and more about just removing unused instruments).

Thank you for your comment. Again, a very good observation. A step in the standardization process is identifying redundant and infrequently used instruments in the traditional trays and removing them from the trays to be put into peel packs so that they do not have to be reprocessed frequently and at the same time they are still available for use. To put it simply, there were instruments in each of the GS, GY, and GO trays that were infrequently used and thus were removed from the SLT, and the rest of the frequently used instruments were combined and thus producing the SLT. As shown in Table 3, the SLT is distinct to the GS, GY, and GO trays. 

There was a lot of dissatisfaction with the customized trays with RNs and MDRs. Why was this?

Thank you for this comment. The dissatisfaction with the traditional customized trays originate from the fact that they are often redundant instruments which makes the set up and instrument selection more tedious. In addition, aside from having redundant instruments, the tray was also frequently missing critical instruments. This was added in Lines 208-210. 

How long after the standardization process was the survey delivered? How was it delivered (onoine,paper etc)? How many people was it sent to vs completed it (% response rate?).

Thank you for your comment. The surveys were distributed 1 month pre- and post- standardization. We have included this in Line 202-203. This survey was given on paper and the response rate was 100%. 

I could not see any data on the reprocessing times for each instrument (as per the method). 

Thank you for this comment. We observed reprocessing times and found minimal variation between instruments, as documented in the paper describing our process.

DISCUSSION

The mathematical focus of this paper ignores many of the operational nuances that may be found in this type of approach.

Thank you for this comment. We have commented on the operational nuances of this intervention and cited the relevant paper in Line 219-220.

“Financial benefits….well documented” – this is a very broad brush. In fact, it’s well documented that the wrong sort of standardization can lead to inefficiencies. Furthermore, standardization and patient-centered care are often in opposition.How do you think your approach would fare with other types of trays e.g. general surgery? These trays often have >100 instruments. What about the trade-offs with

Thank you for this comment. We have described a paper with a similar process of optimizing large trays in other subspecialties in Lines 224-226 and 229-231. Although incorrect standardization can lead to inefficiency and reduced satisfaction, our hybrid model of both computational and clinical optimization describes a novel approach to operative inventory optimization that minimizes the tradeoffs.

How many surgeries required extra instruments (or “peel packs?”)? 

Thank you for this insightful comment. Unfortunately, this data was not collected - we have reflected this shortcoming in the Limitations (Lines 264-265)

One challenge with standard trays is that surgeons who trained differently have different preferences. Did you find that this suited some surgeons more than others; that they all used similar approaches; or that the particular procedures you worked on required a standard set of instruments, with little personal or procedural variation? The cost calculations are well discussed but I think this misses a lot of nuanced implications / discussion points.

Thank you for this insightful comment. We have commented on this challenge at length in the quality improvement paper cited here. The present study primarily describes a quantitative and mathematical technique to standardize trays in a novel fashion. 

Did you calculate the “lifespan” of the instruments? Presumably if they are being turned over faster, they will need to be replaced sooner? 

Thank you for this comment. We address this question in Lines __ (Discussion) where we state that reprocessing and procurement costs were unchanged.

Again, please state over what time period the “savings” (line 220) refer to.

Thank you for this comment. We have specified the time period of savings (annual) in Line 251.

Some trays have rarely used instruments that nevertheless are important to have at the right time. How/are you sure that you didn’t remove instruments like this?

Thank you for this comment. By utilizing a hybrid model of mathematical and clinician review, we mitigated the risk of removing important instruments. Additionally, we ensured that all peel pack instruments were either in the operating room or nearby in the sterile core.

How did the implementation of the new trays happen? Making change in organizations is complex, especially with surgeons who are used to doing this a certain way. Did this present challenges? Overall, the conclusion should consider not just the mathematical calculations, but the safety, quality, and operational implications for this work. Can this approach be used on all trays or only a subset? Are the authors planning more studies in other specialties or procedures or trays? As I’ve said, there’s no consideration (as far as I can tell) for the implications in tray assembly, routine maintenance, or replacement of instruments. 

Thank you for this comment. We feel that the model of standardizing the GS/GY laparoscopic tray was an ideal starting point, due to the familiarity and generalizability of the problem. We have used this approach on instrument trays in other specialties, and will continue applying quantitative strategies to address this common problem. We have expanded on some of the safety, quality, and operational implications of this work in the Discussion (Lines 217-227) and Limitations (Lines 268-276).

Thank you for your time and efforts in enhancing our work.

Sincerely, 

Jay Toor MD MBA

Orthopaedic Surgery Resident Physician

Department of Surgery

University of Toronto

Tel: 1-416-918-9519

Email: jaysinghtoor@gmail.com

---

## [Decision Letter · Decision Letter 2]

6 Oct 2022

Standardization of laparoscopic trays using an inventory optimization model to produce immediate cost savings and efficiency gains

PONE-D-21-17028R2

Dear Dr. Toor,

We’re pleased to inform you that your manuscript has been judged scientifically suitable for publication and will be formally accepted for publication once it meets all outstanding technical requirements.

Kind regards,

Kush Raj Lohani, Master of Surgery

Academic Editor

PLOS ONE

Additional Editor Comments (optional):

"Suggestions - kindly use the full forms in conclusion for - IOM, GS, GY and GO."

I want to congratulate authors on their vision and hard work on this often overlooked topic. I am confident that this manuscript will get a proper recognition in the Surgical field. Readers will get the sense of necessity on proper planning of their instrument trays for enhancing OR efficiency and lowering the avoidable expenses.

Reviewers' comments:

Reviewer's Responses to Questions

**Comments to the Author**

1. If the authors have adequately addressed your comments raised in a previous round of review and you feel that this manuscript is now acceptable for publication, you may indicate that here to bypass the “Comments to the Author” section, enter your conflict of interest statement in the “Confidential to Editor” section, and submit your "Accept" recommendation.

Reviewer #2: All comments have been addressed

Reviewer #3: All comments have been addressed

2. Is the manuscript technically sound, and do the data support the conclusions?

Reviewer #2: Yes

Reviewer #3: Yes

3. Has the statistical analysis been performed appropriately and rigorously? 

Reviewer #2: Yes

Reviewer #3: Yes

4. Have the authors made all data underlying the findings in their manuscript fully available?

Reviewer #2: Yes

Reviewer #3: Yes

5. Is the manuscript presented in an intelligible fashion and written in standard English?

Reviewer #2: Yes

Reviewer #3: Yes

6. Review Comments to the Author

Reviewer #2: This paper is now well written and informative. All the concerns have been well answered and the manuscript was well revised.

Reviewer #3: Thank you for addressing the critique. This is much improved and now an excellent paper that I hope get a good audience.

7. PLOS authors have the option to publish the peer review history of their article (what does this mean?). If published, this will include your full peer review and any attached files.

Reviewer #2: **Yes: **Tuerhongjiang Tuxun

Reviewer #3: No

---

## [Editor Report · Acceptance letter]

17 Nov 2022

PONE-D-21-17028R2 

Standardization of laparoscopic trays using an inventory optimization model to produce immediate cost savings and efficiency gains 

Dear Dr. Toor:

I'm pleased to inform you that your manuscript has been deemed suitable for publication in PLOS ONE. Congratulations! Your manuscript is now with our production department. 

Kind regards, 

on behalf of

Dr. Kush Raj Lohani 

Academic Editor

PLOS ONE